LETTER TO THE EDITOR

# Lower Incidence of Parvovirus-B19 Infections in Dutch Blood Donors during SARS-CoV-2 Pandemic

M. W. Molenaar-de Backer,[a] B. M. Hogema,[a,b] M. H. Koppelman,[c] T. J. van de Laar,[b] E. Slot,[d] H. L. Zaaijer[b,e]

[a]Department of Virology and MAT Services, Sanquin Diagnostiek, Amsterdam, The Netherlands
[b]Department of Donor Medicine Research Laboratory of Blood-Borne Infections, Sanquin Research, Amsterdam, The Netherlands
[c]National Screening Laboratory, Sanquin Research, Amsterdam, The Netherlands
[d]Department of Medical Affairs, Sanquin Corporate Staff, Amsterdam, The Netherlands
[e]Department of Clinical Virology, Amsterdam University Medical Centre, Amsterdam, The Netherlands

**KEYWORDS** parvovirus B19, SARS-CoV-2, blood donor

During the SARS-CoV-2 pandemic several studies reported a lower incidence of respiratory infections other than COVID-19 compared to previous years, but bias due to changes in uptake and capacity of non-COVID care cannot be excluded (1, 2). The demand for blood did not decrease during the SARS-CoV-2 outbreak, and Dutch donors responded en masse to the request to continue donating blood. Therefore, the observed incidence of infectious diseases with routes of transmission similar to SARS-CoV-2 among blood donors may not suffer from these potential testing biases. In fact, the number of blood donations per month in 2020 and 2021 was comparable to the years before (719,211 donations in 2020 compared to min-max range of 682,315 [2019] to 721,183 [2013]). Parvovirus B19 (B19V), a member of genus *Erythroparvovirus*, causes the childhood disease erythema infectiosum, or fifth disease, but many B19V infections remain asymptomatic, especially in immunocompetent adults. Transmission of B19V occurs primarily via the respiratory route but also via transfusion of blood components and plasma products (3, 4). In the past, transmission of B19V by plasma-derived medicines occurred frequently because nonenveloped viruses are difficult to inactivate by heat or detergents during the plasma fractionation process. To comply with the European guidelines that plasma pools for the production of certain plasma-derived products may not exceed 10,000 IU/ml B19V DNA Sanquin Blood Supply Foundation routinely screens donated plasma for fractionation for B19V to exclude donations with a high B19V load ($>1.2 \times 10^6$ B19V DNA IU/ml) before fractionation.

From 1 January 2013 to 30 June 2021, Sanquin screened 5,974,061 plasma donations for B19V DNA, as described previously (5). The standard test pool size was 480, but donations used for production of solvent-detergent-treated plasma were screened in pools of 96. When a pool of 480 or 96 donations contained an unacceptable level of B19V DNA ($>5,000$ IU/ml or $>1,000$ IU/ml for pool size of 480 or 96 donations, respectively), the pool was deconstructed to identify the donation(s) containing a high load of B19V DNA. During the study period, 200 donations were identified which contained $>1.2 \times 10^6$ IU/ml of B19V DNA. As shown in our previous study (6), high-level viremia is a hallmark of acute B19V infection, while persistent low levels of B19V DNA ($>1 \times 10^5$ IU/ml) are often associated with the detection of viral remnants. Figure 1 shows the number of highly viremic donations per 10,000 donations for each month in 2013 to 2020. Following a seasonal pattern, most B19V infections occur from December to July, with a peak around April. January 2013 and March 2017 showed the highest incidence, with 1 highly viremic donation per 6,697 and 6,572 donations, respectively. The lowest B19V incidences typically occur in October and November. Interestingly, in 2020 very few donations with B19V DNA loads of $>1.2 \times 10^6$ IU/ml were detected; in 2020 only 4 B19V viremic

Address correspondence to M. W. Molenaar-de Backer, m.molenaar@sanquin.nl.

The well defined and unbiased Dutch blood donor population showed that there was a significant reduction of silent Parvovirus B19 infections due to public health measures taken to limit the spread of SARS-CoV-2.

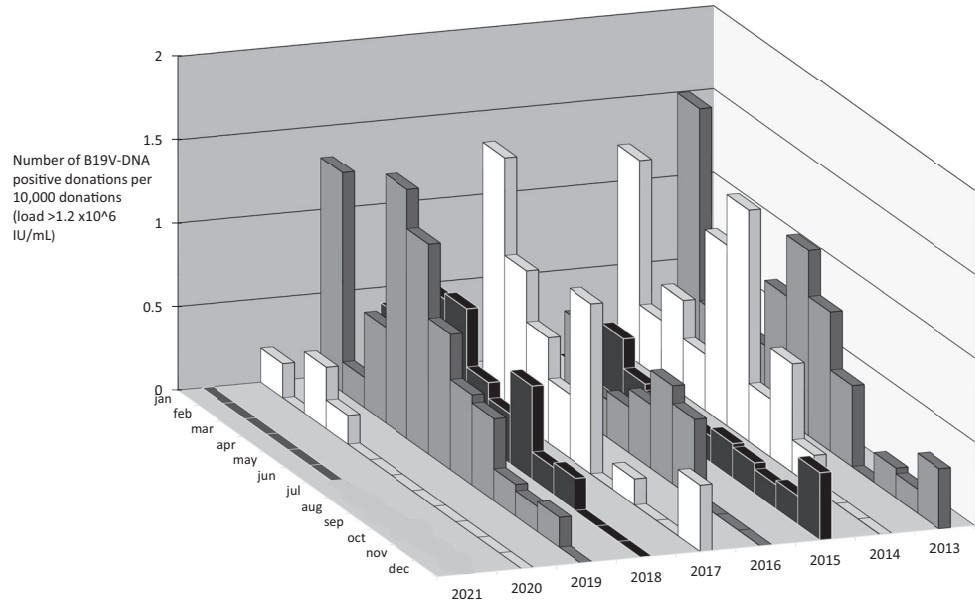

**FIG 1** The number of parvovirus B19 highly viremic donations per month per 10,000 donations (donations with a B19V DNA load of >1.2 × 10^6 IU/ml).

donations were identified, and 0 were identified in the first 6 months of 2021, compared to 13 (2015) to 41 (2019) donations with loads of >1.2 × 10^6 IU/ml in the years before (Fig. 1). The number of donations with high B19V DNA loads during the first 4 months of 2020 were comparable to the first months of low-prevalence B19V years 2015 and 2016. However, from May 2020 until 30 June 2021, zero B19V highly viremic blood donations were detected. This sharp decrease in highly viremic donations coincides with, and probably is causally related to, the COVID-19 epidemic. From 15 March to 11 May 2020 and from 15 December 2000 to 15 March 2021, all child daycare facilities, schools, cafes, restaurants, and sports clubs were closed in the Netherlands, and people had to work from home if possible. Additional measures including 1.5-m distancing, the obligation to wear face masks in public areas, more emphasis on hand hygiene, and limiting the number of social contacts were also practiced on a large scale between and after the lockdowns. All measures aimed to reduce the spread of SARS-CoV-2 but may have had similar effects on the spread of B19V in The Netherlands. For B19V, closing daycare facilities and schools probably prevented many transmission events between (young) children and subsequently the onward spread to their parents (potential blood donors).

On top of the seasonal cycle, symptomatic B19V infections are known to peak every 3 to 4 years (7). This interepidemic cycle is less obvious in viremic blood donors (see Fig. 1). As previously described, the absence of this 3- to 4-year cycle can be related to the fact that infected blood donors are asymptomatic and feel well enough to donate blood (8); the number of notified B19V cases is often limited to children with symptomatic fifth disease.

Our data showed a significant reduction of silent parvovirus B19 infections, from 4.0/100,000 donations in 2013 to 2019 to 0.56/100,000 and 0.00/100,00 in 2020 and 2021, respectively. The prevalence in 2020 was significantly lower than in previous years, including the year with the lowest prevalence, 2015 (P = 0.04 for 2015 versus 2020) (9). When comparing the 1.5-year period (January 2015 to June 2016) with the lowest B19 prevalence with the most recent 1.5-year period (January 2020 to June 2021), this difference is even more pronounced (P = 0.0014). Since our data set does not suffer from bias caused by changes in the uptake of health care or a decreased testing capacity for infections other than SARS-CoV-2, our findings confirm that public

health measures taken to limit the spread of SARS-CoV-2 reduced the transmission of other respiratory infections, such as B19V.

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
