## [Reviewer comments · Microbiology Spectrum]

**Microbiology
Spectrum**

Lower incidence of parvovirus-B19 infections in Dutch blood donors during SARS-CoV-2 pandemic.

Marijke Molenaar-de Backer, Boris Hogema, Marco Koppelman, Thijs van de laar, Ed Slot, and Hans Zaaijer

Corresponding Author(s): Marijke Molenaar-de Backer, Sanquin Diagnostiek B.V.

Review Timeline:

Submission Date:	May 17, 2021
Editorial Decision:	June 18, 2021
Revision Received:	August 1, 2021
Accepted:	August 9, 2021

Editor: Rosemary She

Reviewer(s): Disclosure of reviewer identity is with reference to reviewer comments included in decision letter(s). The following individuals involved in review of your submission have agreed to reveal their identity: Ping Fu (Reviewer #3)

Transaction Report:

DOI: <https://doi.org/10.1128/Spectrum.00253-21>

June 18, 2021

Dr. Marijke Molenaar-de Backer
Sanquin Diagnostiek B.V.
Virology and MAT services
Plesmanlaan 125
Amsterdam 1066CX
Netherlands

Re: Spectrum00253-21 (Lower incidence of parvovirus-B19 infections in Dutch blood donors during SARS-CoV-2 pandemic.)

Dear Dr. Marijke Molenaar-de Backer:

Thank you for submitting your manuscript to Microbiology Spectrum. The reviewers found interest in your work and we will be happy to consider a modified submission provided you address the reviewer comments below.

When submitting the revised version of your paper, please provide (1) point-by-point responses to the issues raised by the reviewers as file type "Response to Reviewers," not in your cover letter, and (2) a PDF file that indicates the changes from the original submission (by highlighting or underlining the changes) as file type "Marked Up Manuscript - For Review Only". Please use this link to submit your revised manuscript - we strongly recommend that you submit your paper within the next 60 days or reach out to me. Detailed information on submitting your revised paper are below.

Link Not Available

Sincerely,

Rosemary She

Journals Department
Reviewer comments:

Reviewer #1 (Comments for the Author):

The manuscript 00253-21 describes briefly the detection of human parvovirus B19 (B19V) among blood donors in the Netherlands during 2020, as compared to previous years 2013-2019. The authors observe a sharp decrease in the detection rate of highly B19V viremic donations during 2020, which they relate to the lock down measures implemented in order to prevent transmission of SARS-CoV-2. Being a virus which uses the respiratory airway as a main route of transmission, it can be assumed that social distancing, use of face masks, improved hygiene -especially hands- hygiene and other measures, such as closing daycare facilities and schools, can affect the virus prevalence among adults (donors). It should be considered that a certain proportion of detection is associated to persistent infection. In addition, the authors mention that closure measures were implemented from March 15th until May 11th. However the virus is already detected at a much lower rate at the beginning of the year (January, March, April; a statistical test can be applied). Interestingly, it has been shown that lock down measures have indeed affected (reduced) the circulation of other enveloped respiratory viruses, such as Flu, RSV and human metapneumovirus, but not so in the case of non enveloped viruses like rhinovirus, adenovirus and human bocavirus.

Please check line 31, page 2.

Reviewer #2 (Comments for the Author):

This is an interesting observation, in particular the magnitude of the effect on the prevalence of acute B19V infections is remarkable. This effect clearly exceeds the well known four yearly oscillation pattern, as a very low figure, though this oscillation is not specifically considered.

Some suggestions:

- (urgent) to use throughout the manuscript the ICTV advised abbreviation B19V for this virus instead of PV B19. Also good to refer to its genus Erythroparvovirus.
https://talk.ictvonline.org/ictv-reports/ictv_online_report/ssdna-viruses/w/parvoviridae/1044/genus-erythroparvovirus

- to mention the (about) four year oscillation as a reference to the observed pattern: as 2019 was the highest recorded prevalence, 2020 was expected to be low anyway.

That should be taken into account, even as the 2020 figure is lower than ever. To be considered is whether Fourier analysis could support that this value indeed exceeds any expected periodicity but it clearly does at first sight.

- Minor points.

It is stated about a series of measures that these reduced the spread of SARS-CoV-2 "impressively". However, this has NOT been demonstrated for each of these measures so this impression lacks formal evidence and should better be formulated with some caution. The measures were certainly aimed at SARS-CoV-2 transmission, the extent of their effectiveness is largely unclear.

- the precise cut off values for the detectable viral load is not entirely clear with the two pool sizes, please simply provide these.

- line 31 lacks some reference source

Reviewer #3 (Comments for the Author):

1. Appropriate statistical tests were needed to applied in this papaer, even that "4.0/100,000 donations in 2013 - 2019 to 0.6/100,000 donations in 2020" seems the difference is obvious.
2. In the first sentence, "several studies reported a lower incidence of respiratory infections other than COVID-19, as compared to previous years" ,such as ???. I also suggest that references should be cited.

Staff Comments:

Preparing Revision Guidelines

For complete guidelines on revision requirements, please see the Instructions to Authors at [link to page]. **Submissions of a paper that does not conform to Microbiology Spectrum guidelines will delay acceptance of your manuscript.**

Please return the manuscript within 60 days; if you cannot complete the modification within this time period, please contact me. If you do not wish to modify the manuscript and prefer to submit it to another journal, please notify me of your decision immediately so that the manuscript may be formally withdrawn from consideration by Microbiology Spectrum.

If you would like to submit an image for consideration as the Featured Image for an issue, please contact Spectrum staff.

The manuscript 00253-21 describes briefly the detection of human parvovirus B19 (B19V) among blood donors in the Netherlands during 2020, as compared to previous years 2013-2019. The authors observe a sharp decrease in the detection rate of highly B19V viremic donations during 2020, which they relate to the lock down measures implemented in order to prevent transmission of SARS-CoV-2.

Being a virus which uses the respiratory airway as a main route of transmission, it can be assumed that social distancing, use of face masks, improved hygiene -especially hands- hygiene and other measures, such as closing daycare facilities and schools, can affect the virus prevalence among adults (donors). It should be considered that a certain proportion of detection is associated to persistent infection. In addition, the authors mention that closure measures were implemented from March 15th until May 11th. However the virus is already detected at a much lower rate at the beginning of the year (January, March, April; a statistical test can be applied). Interestingly, it has been shown that lock down measures have indeed affected (reduced) the circulation of other enveloped respiratory viruses, such as Flu, RSV and human metapneumovirus, but not so in the case of non enveloped viruses like rhinovirus, adenovirus and human bocavirus.

Please check line 31, page 2.

Dear editor

Thank you for giving us the opportunity to revise our manuscript Spectrum 00253-21 entitled " Lower incidence of parvovirus-B19 infections in Dutch blood donors during SARS-CoV-2 pandemic.". We thank the reviewers for their valuable suggestions.

We have revised the manuscript according to the reviewers suggestions. We have submitted a new version of the manuscript which shows the sections and fragments that we changed in yellow.

During the period of manuscript writing and submission additional data for Dec 2020 until Jun 2021 has become available and this data was added to the manuscript because we feel it strengthens the conclusion.

On behalf of all authors of Spectrum 00253-21,

Marijke Molenaar-de Backer, PhD

-----///-----

Reviewer 1

1. *It should be considered that a certain proportion of detection is associated to persistent infection.*

Previous results from our group have shown that donations with a B19V load above 1×10^5 IU/mL are acute infections and that persistent 'infections' were associated with B19V loads below 1×10^5 IU/mL and these were characterized by presence of DNA remnants rather than virions (see Molenaar-de Backer et al 2016 J. Clinical Virology) Thus, the donations with loads $>1.2 \times 10^6$ IU/mL are not associated with persistent infection. We have included a sentence on this phenomenon.

"As shown by our previous study [6], high-level viremia is a hallmark of acute B19V infection, while persistent low levels of B19V DNA ($>1 \times 10^5$ IU/mL) are often associated with the detection of viral remnants."

2. *In addition, the authors mention that closure measures were implemented from March 15th until May 11th. However the virus is already detected at a much lower rate at the beginning of the year (January, March, April; a statistical test can be applied).*

In 2015 and 2016 there were also no donations detected with B19V loads $>1.2 \times 10^6$ IU/mL in the first months of the year, so during the first months there was no significant difference between the prevalence in 2020 and other years. It appears that 2020 started as a "normal" low prevalent B19V year, however the COVID-19 measures have reduced the number of donations with high B19V loads strongly, since until 30th of June 2021 only 4 high B19V load donations were detected. We have included some clarification on this in the manuscript:

"The number of donations with high B19V DNA loads during the first 4 months of 2020 were comparable to the first months of low prevalent B19V years 2015 and 2016. However, from May 2020 until 30th of June 2021 zero B19V highly viremic blood donations were detected."

3. *Check line 31 page 2*

Thank you for indicating this error. The reference to figure 1 was incorrect and yielded a reference error. This is corrected in the revised manuscript.

Reviewer 2

1. *To use throughout the manuscript the ICTV advised abbreviation B19V for this virus instead of PV B19. Also good to refer to its genus Erythroparvovirus.*

Thank you for indicating the correct ICTV abbreviation. We have changed the abbreviation from PV B19 to B19V.

2. *To mention the (about) four year oscillation as a reference to the observed pattern: as 2019 was the highest recorded prevalence, 2020 was expected to be low anyway. That should be taken into account, even as the 2020 figure is lower than ever. To be considered is whether Fourier analysis could support that this value indeed exceeds any expected periodicity but it clearly does at first sight.*

Thank you for this suggestion. We have included a remark on the 4-year oscillation pattern. Since previous studies have shown that the B19V infections in blood donors are asymptomatic and do not follow the 3-4 year oscillation pattern observed in children with fifth disease we decided to perform caldoff test as described by Janssen M.P. et al. 2009 (Monitoring viral incidence rates: tools for the implementation of European Union regulations, Vox Sang 2009;96: 298-30) comparing 2020 with the year with the lowest prevalence, 2015. In addition we have added data from Dec 2020 and 1st Jan – 30th of June 2021 and compared the 1.5 year period from Jan 2020 – Jun 2021 with period Jan 2015- Jun 2016 with the same test.

3. *It is stated about a series of measures that these reduced the spread of SARS-CoV-2 "impressively". However, this has NOT been demonstrated for each of these measures so this impression lacks formal evidence and should better be formulated with some caution. The measures were certainly aimed at SARS-CoV-2 transmission, the extent of their effectiveness is largely unclear.*

Thank you for this suggestion, we have changed the wording in the concerning paragraph to

“All measures aimed to reduce the spread of SARS-CoV-2, but may have had similar effects on the spread of B19V in the Netherlands. For B19V, closing daycare facilities and schools probably prevented many transmission events between (young) children and subsequently the onward spread to their parents (potential blood donors).”

4. *The precise cut off values for the detectable viral load is not entirely clear with the two pool sizes, please simply provide these.*

We have included the cut off values for the pools of 480 and 96 donations, by inserting “(>5,000 IU/mL or >1,000 IU/mL for pool size of 480 or 96 donations, respectively)”.

5. *line 31 lacks some reference source*

Thank you for indicating this error. The reference to figure 1 was incorrect and yielded a reference error. This is corrected in the revised manuscript.

Reviewer 3

1. *Appropriate statistical tests were needed to applied in this papaer, even that "4.0/100,000 donations in 2013 - 2019 to 0.6/100,000 donations in 2020" seems the difference is obvious.*

We have included statistical caldoff test as described by Janssen M.P. et al. 2009 (Monitoring viral incidence rates: tools for the implementation of European Union regulations, Vox Sang 2009;96: 298-

30) to calculate the p-values for difference between 2020 and 2015 and also the last 1.5 year and the 1.5 year period with the lowest B19V prevalence (Jan 2015 – Jun 2016).

2. *In the first sentence, "several studies reported a lower incidence of respiratory infections other than COVID-19, as compared to previous years" ,such as ??? . I also suggest that references should been cited.*

Thank you for this remark, we have added references to two of these studies.

August 9, 2021

Dr. Marijke Molenaar-de Backer
Sanquin Diagnostiek B.V.
Virology and MAT services
Plesmanlaan 125
Amsterdam 1066CX
Netherlands

Re: Spectrum00253-21R1 (Lower incidence of parvovirus-B19 infections in Dutch blood donors during SARS-CoV-2 pandemic.)

Dear Dr. Marijke Molenaar-de Backer:

Your manuscript has been accepted, and I am forwarding it to the ASM Journals Department for publication. You will be notified when your proofs are ready to be viewed.

Sincerely,

Rosemary She
Editor, Microbiology Spectrum
